# Concept Learning for Interpretable Multi-Agent Reinforcement Learning

**Renos Zabounidis**,* **Joseph Campbell**,* **Simon Stepputtis, Dana Hughes, Katia Sycara**

Carnegie Mellon University
{renosz, jacampbe, sstepput, danahugh, sycara} @andrew.cmu.edu

**Abstract:** Multi-agent robotic systems are increasingly operating in real-world environments in close proximity to humans, yet are largely controlled by policy models with inscrutable deep neural network representations. We introduce a method for incorporating interpretable concepts from a domain expert into models trained through multi-agent reinforcement learning, by requiring the model to first predict such concepts then utilize them for decision making. This allows an expert to both reason about the resulting concept policy models in terms of these high-level concepts at run-time, as well as intervene and correct mispredictions to improve performance. We show that this yields improved interpretability and training stability, with benefits to policy performance and sample efficiency in a simulated and real-world cooperative-competitive multi-agent game.

**Keywords:** Multi-Agent Reinforcement Learning, Interpretable Machine Learning

## 1 Introduction

With burgeoning adoption in fields such as autonomous driving, service robotics, and healthcare, multi-agent robotic systems are increasingly operating in real-world environments. The actions of these systems have a tangible and significant impact, particularly so when operating in close proximity to humans. While we expect such systems to exhibit safe and accurate behavior, errors are inevitable, and in such circumstances it is vitally important that the agents are able to explain their behavior to human operators. Operators can then ascertain whether the agent is operating erroneously – thus requiring intervention – or correctly but in a non-obvious manner.

However, state-of-the-art multi-agent systems are often controlled by deep neural network models trained with reinforcement learning techniques [1]. While these methods have shown great ability to generate effective and generalizable models, they do so at the expense of interpretability, and the models often remain inscrutable to human operators [2]. This poses a significant risk, especially in end-to-end models, where it is not clear what information has been extracted from raw observations in order to make a policy decision. Domain experts often reason about agent behavior in terms of high-level *concepts* such as the presence of an obstacle – e.g., "the robot encountered an obstacle and subsequently changed direction". However, standard end-to-end models provide no mechanism for this sort of reasoning, let alone the ability to intervene and correct the model when it is wrong – e.g., "the robot should have detected an obstacle but it didn't". Such a mechanism is particularly important for robotic systems where we often encounter shifts in data distributions, such as when transferring policies from simulated environments to the real world, leading to model errors. These errors are exacerbated in multi-agent systems, where errors in each individual agent are compounded and produce large errors in environment dynamics.

In this paper, we propose a method for learning interpretable policies – concept policy models – for multi-agent reinforcement learning (MARL). Our approach is predicated on the insight that we can leverage domain knowledge from an expert in order to regularize the model and influence what information is encoded from observations. We organize this domain knowledge into a set of interpretable concepts and enforce the constraint that the model is able to predict these concepts from observations, after which the concepts are used to predict policy actions. Concepts are semantically

---

*Equal contribution

6th Conference on Robot Learning (CoRL 2022), Auckland, New Zealand.

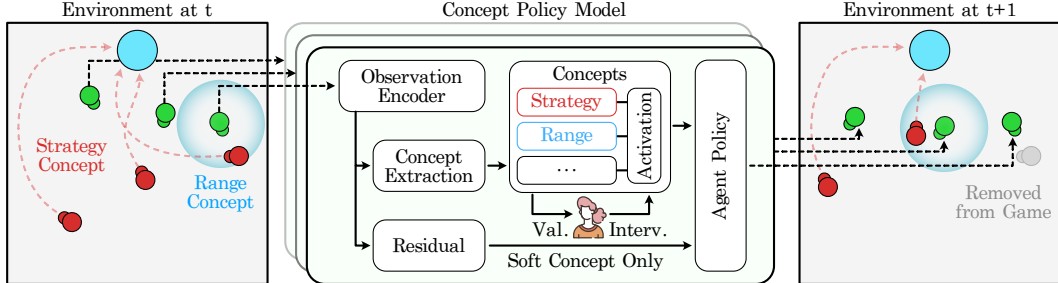

Figure 1: Concept policy models predict a set of interpretable concepts from observations, which are then used along with an (optional) residual to predict a policy action. A domain expert may intervene and provide corrective concept values to the policy if mis-predicted.

meaningful labels that can be extracted from observations, such as the presence of a concrete or abstract feature in an observation, e.g., the existence of an obstacle or the intention of a human. Crucially, we find that the regularization imposed by the concept information helps stabilize the training process, and as a result leads to improved performance and sample efficiency.

A typical end-to-end neural network policy model maps observations to actions [3]. Our approach inserts an intermediate *concept* layer, as shown in Fig. 1 which is required to predict concepts from observations. While this yields an interpretable model [4], it also imposes the assumption that the set of concepts are sufficient for policy inference. To ease this constraint, we introduce a scalable *residual* layer which passes additional information to the subsequent policy layers while ensuring it remains decorrelated with the concepts. We posit that the interpretability of the model is proportional to the capacity of the residual layer; intuitively, the more residual information available, the less the model may rely on the concepts. We show that this can result in a trade-off between interpretability and accuracy, given the expressivity of the concepts. Our contributions are as follows, we

- Introduce a method for learning concept policy models in MARL utilizing expert domain knowledge, enabling the expert to reason about policy behavior in terms of high-level concepts and improving accuracy, sample efficiency, and training stability.

- Develop two specific formulations based on this method: soft-concept models and hard-concept models, and empirically show the trade-off between accuracy and interpretability.

- Formulate an intervention methodology and show how this can be used to offset model errors in general and in transfer learning (sim-to-real) scenarios.

- Empirically show that our proposed approach produces interpretable, intervenable MARL policy models which exceed the accuracy of baseline MARL policies in a simulated and real-world game of "tag", between two teams of 2, 3, and 5 robots each.

## 2   Related Work

**Interpretability in supervised learning**: Intepretability has been extensively studied within the field of supervised learning [5], which can be largely grouped into two categories: the explicit creation of an intrinsically interpretable model, or the post-hoc transformation of an uninterpretable model to an interpretable one. The former case typically revolves around considering interpretable classes of models – decision trees [6, 7], linear models [8, 9], or rule-based methods [10, 11] for example – and developing algorithms for these models. In the latter case, uninterpretable models are either transformed into interpretable ones [12, 13, 14], or interpretable models are extracted from an uninterpretable model for the purpose of explaining a model's decision rationale [15, 16].

Concept models often fall into the transformation case, and have been studied within the context of transforming a set of uninterpretable feature embeddings into a set of interpretable concepts [4, 17, 18]. A recent approach [19] similarly uses concepts, but rather than directly predicting such concepts it attempts to align the internal model representation to coincide with them.

**Interpretability in reinforcement learning**: As in supervised learning, interpretability for reinforcement learning largely falls into the two categories of intrinsically interpretable models and post-hoc

transformations. However, there is an additional line of work in which methods are devised to explain aspects particular to the Markov decision process model employed by RL methods. Some approaches have focused on interpretable representation learning [20, 21] and hierarchical decompositions [22], while others have opted to tackle MDP-specific explanations such as an interpretable reward signal [23] or action explanation [24, 25, 26]. However, to the best of our knowledge, concept-related methods have not yet been explored in an RL setting, let alone MARL.

## 3  Preliminaries

**Multi-Agent Reinforcement Learning**: We model the MARL problem as a decentralized partially observable Markov decision process (Dec-POMDP) [27]. A Dec-POMDP is defined as a tuple $\langle \mathcal{S}, \mathcal{U}, P, R, \mathcal{Z}, O, n, \gamma \rangle$ in which $\mathcal{S}$ is the state space, $\mathcal{U}$ shared action space, $P$ the state transition function, $R$ the shared reward function, $\mathcal{Z}$ the observation space, $O$ the observation function, $n$ the number of agents, and $\gamma$ the discount factor. For a given time step, the environment has a state $\mathbf{s} \in \mathcal{S}$ and each agent $a \in \{a_0, \ldots, a_n\}$ samples a partial observation $\mathbf{z}_a \in \mathcal{Z}$ according to the observation function $O(s, a) \in \mathcal{Z}$. The agents simultaneously sample an action $\mathbf{u}_a \in \mathcal{U}$ inducing a state transition according to $P(\mathbf{s}'|\mathbf{s}, \mathbf{u}) \in [0, 1]$. Each agent receives a reward $r_a$ according to the shared reward function $R(\mathbf{s}_a, \mathbf{u}_a) \in \mathbb{R}$ with a discount factor $\gamma \in [0, 1]$. We follow a centralized training and decentralized execution approach (CTDE), thus learning a central policy $\pi_\theta(\mathbf{u}_a|\mathbf{z}_a) \in [0, 1]$ parameterized by $\theta$ by maximizing the discounted expected cumulative reward: $\mathbb{E}_t[\sum_t \gamma^t R(\mathbf{s}_a, \mathbf{u}_a)]$.

**Multi-Agent Proximal Policy Optimization**: Multi-Agent Proximal Policy Optimization (MAPPO) [28] is a straightforward extension to standard PPO [29] under the CTDE assumption in which we learn a single actor, $\pi_\theta$, and a single critic, $V_\phi$, parameterized by $\theta$ and $\phi$ respectively. When sampling from the environment, each agent executes the same learned policy with their individual observations and actions. As with all policy gradient methods, PPO seeks to compute the policy gradient by differentiating the following objective function:

$$L(\theta) = \hat{\mathbb{E}}_t[\log \pi_\theta(\mathbf{u}_a|\mathbf{z}_a)\hat{A}_t], \tag{1}$$

where $\hat{A}$ is the estimated advantage function. PPO extends this objective function by adaptively clipping the update gradient and applying an entropy bonus to the policy to encourage exploration. If the value function and policy function share parameters, i.e., $\theta = \phi$, then the objective function must also include the value function loss.

## 4  Concept Policy Models

We propose a method for learning concept policy models, which integrates domain knowledge from an expert in the form of concepts into a neural network policy model. These concepts are intended to serve two purposes: they are useful predictors for the desired policy behavior and as such allow an expert to reason about the policy in terms of high-level concepts, and mispredicted concepts can be corrected at run-time by the expert in order to induce correct behavior. The expert – hereafter referred to as an oracle – provides an oracle function $V(\cdot)$ which can be used to predict a ground truth concept vector of size $j$ given an observation from an agent $a$, $\mathbf{v} = V(\mathbf{z}_a)$ where $\mathbf{v} \in \mathbb{R}^j$. Concepts may be either continuous or discrete, and represent interpretable features which are assumed to be relevant to the task at hand. As an example, let us consider a cooperative-competitive multi-agent game in which two teams of agents play a game of "tag" during which one side must prevent the other from reaching a specific location. In this game, an expert might identify specific features such as the location of the nearest enemy, or the opposing team's strategy as a concept.

### 4.1  Policy Concept and Residual Layers

We integrate this concept information into an end-to-end neural network policy $\pi_\theta(\mathbf{u}_a|\mathbf{z}_a)$ which predicts the probability for agent $a$ taking action $\mathbf{u}_a$ given observation $\mathbf{z}_a$ and parameters $\theta$. This is accomplished by inserting an intermediate layer $c_{\theta c}(\cdot)$ into the network to predict the concept vector, dividing the network into two parts: $\pi_{\theta 1}^{(1)}(\cdot)$ representing the portion of the network before the new layer, and $\pi_{\theta 2}^{(2)}(\cdot)$ representing the portion of the network after the new layer, such that

$$\pi(\mathbf{u}_a|\mathbf{z}_a) = \pi_{\theta 2}^{(2)}(c(\mathbf{x}) + r(\mathbf{x})) \quad \text{where} \quad \mathbf{x} = \pi_{\theta 1}^{(1)}(\mathbf{z}_a) \tag{2}$$

and $r(\cdot)$ is a residual layer of size $k$ designed to pass through non-concept information and the concept layer acts as a concept predictor, such that $\hat{v} = c(\mathbf{x})$. In our proposed concept policy model,

$\pi_{\theta 1}^{(1)} : \mathbb{R}^{|\mathbf{z}|} \to \mathbb{R}^h$ acts as a feature encoder mapping an observation to a feature embedding. The newly inserted concept layer serves as a bottleneck such that $c(\cdot) : \mathbb{R}^h \to \mathbb{R}^j$ maps the feature embedding to a concept vector, while the residual layer $r(\cdot) : \mathbb{R}^h \to \mathbb{R}^k$ maps the feature embedding to a residual vector. The final policy layer $\pi_{\theta 2}^{(2)}(\cdot) : \mathbb{R}^{j+k} \to \mathbb{R}^{|\mathbf{u}|}$ maps the aggregated concept and residual vectors to a policy action. We train the concept layer $c(\cdot)$ by imposing an additional auxiliary loss $L^c(\theta)$ in the objective function optimized by MAPPO:

$$L(\theta) = \hat{\mathbb{E}}_t[\log \pi_\theta(\mathbf{u}_a | \mathbf{z}_a)\hat{A}_t] - L^c(\theta) \quad \text{where,}$$

$$L^c(\theta) = \sum_{i=0}^{j} L_i^c(\theta) \quad \text{and} \quad L_i^c(\theta) = \begin{cases} \text{FL}(v_i, \hat{v}_i) & \text{if discrete} \\ \text{MSE}(v_i, \hat{v}_i) & \text{if continuous.} \end{cases} \tag{3}$$

This loss is summed over each concept: mean squared error (MSE) is used for continuous concepts, and focal loss (FL) [30] for discrete concepts. The focal loss is a cross-entropy variant designed for class imbalanced situations which are likely to occur in our concept setting, as some concepts may be significantly rarer than others. In our above example with the strategy concepts, some strategies may be much less likely to occur than others, for instance. Note that for multi-class discrete concepts, a single abstract concept may consist of multiple nodes, and we refer to this as a *concept group*. In the strategy case, suppose an agent team may only execute one of strategy $A$, $B$, or $C$ at a time, thus these three concepts represent a single concept group and so when we pass the discrete concepts through a softmax activation in order to calculate the focal loss we do so in a group-wise manner.

The goal of the residual layer is to pass through information from $\pi_{\theta 1}^{(1)}(\cdot)$ that is not captured by the concept vector. Without the residual, the concept vector must sufficiently represent the observation so that $\pi_{\theta 2}^{(2)}$ accurately infer the agent's action from concepts alone (a strict assumption in practice). We define two concept policy model variants: **hard concept policy models** which contain no residual ($k = 0$), and **soft concept policy models** which do ($k > 0$).

By examining the concept layer activations, an oracle can query the predicted concepts $\hat{v}$ and understand what concepts the policy model used for prediction. However, we conjecture that there is an inherent trade-off between the size of the residual layer $k$ and the interpretability of these activations. While a full interpretability analysis is outside the scope of this work, we posit that the greater the residual dimension, the less that $\pi_{\theta 2}^{(2)}(\cdot)$ must rely on the concept vector, i.e., there is a larger amount of non-concept information on which to base its prediction – which follows that $k$ is inversely proportional to interpretability.

## 4.2 Concept and Residual Whitening

In order to constrain the residual $r(\cdot)$ such that it does not encode information related to the concepts, we decorrelate the neuron activation vectors via whitening. Given a matrix $\mathbf{X} \in \mathbb{R}^{b \times j+k}$ consisting of the activations from the concatenated concept and residual vectors over a mini-batch of $b$ samples, we aim to produce a whitened matrix $\mathbf{X}'$ with ZCA whitening via iterative normalization [31]

$$\mathbf{X}' = \mathbf{D}\Lambda^{-\frac{1}{2}}\mathbf{D}^T(\mathbf{X} - \mu_x) \tag{4}$$

where $\mathbf{D}$ and $\Lambda$ are the eigenvectors and eigenvalues of $\mathbf{X}$ respectively. Iterative normalization uses an iterative optimization technique to incrementally whiten the matrix $\mathbf{X}$, where the hyperparameter $T$ dictates the number of optimization iterations. This gives us the flexibility of only partially decorrelating the residual and whitening layers, if desired, by setting $T$ to a smaller value, e.g., $T = 2$. In practice, we find that performing fewer iterations is often necessary to stabilize the training process, as a higher $T$ tends to increase the stochastic normalization disturbance and leads to reduced training efficiency [31], which is particularly noticeable in a MARL setting. At each training iteration, we first perform whitening then backpropagate our computed gradients, thus allowing us to decorrelate the concept and residual layers without requiring an additional optimization step as in prior work [19].

## 4.3 Policy Intervention

In addition to querying the predicted concepts $\hat{v}$, an oracle may also decide to intervene when these predictions are incorrect. This can be achieved by explicitly overwriting the concept layer node activations (or softmax activations for discrete concepts) with the appropriate values. We denote the modified concepts as $\bar{v}$ which leads to the following intervened concept policy model:

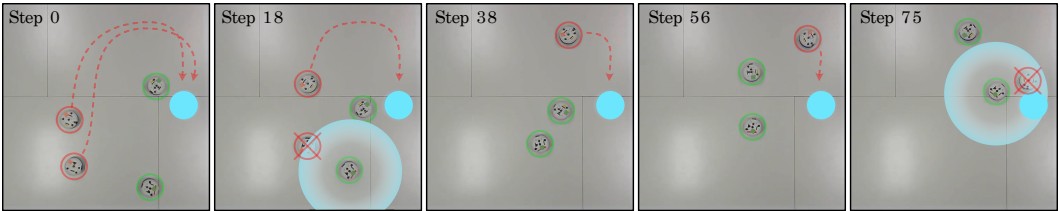

Figure 2: A sequence of steps from a single episode during a policy execution in the real-world. The blue circle is the attacking team's (red) goal while the defending team (green) attempts to stop them.

$\pi_\theta(\mathbf{u}_a | \mathbf{z}_a, \bar{v}) = \pi_{\theta 2}^{(2)}(\bar{v} + r(\mathbf{x}))$. This policy intervention corrects prediction errors in the feature encoder $\pi_{\theta 1}^{(1)}(\cdot)$. We find that these sorts of errors are particularly prevalent when transferring robot policies from simulation to the real world due to different observation distributions, and show that policy interventions are effective at reducing such errors in Sec. 6. Similar to interpretability, we hypothesize that intervention effectiveness is inversely proportional to the size of the residual $k$; we cannot intervene on the residual layer activations so any resulting errors will persist. While our empirical results hint in this direction, we save a full exploration for future work.

## 5  Experimental Setup

We show that our proposed concept policy models achieve high policy success rates and concept accuracy, in addition to improved training stability and sample efficiency, over standard MARL models in a cooperative-competitive multi-agent game of "tag" previously described in Sec. 4. We empirically analyze our approach in both simulated and real-world versions of this game, and explore its strengths and weaknesses especially with respect to interventions and sim-to-real transfer in Sec. 6.

### 5.1  Tag Game

In our game shown in Fig. 2, two equally sized teams of agents compete with each other in which one team attempts to reach a specified goal location while the other team defends it and attempts to keep them away, which we refer to as the attacking and defending team respectively. To allow for complex behaviors and strategy, an agent from each team may "tag" an agent from the opposing team as long as it lies within a given proximity and is facing the opposing agent, removing the tagged agent from play. The attacking team wins if any agent is able to reach the goal location, while the defending team wins if the attacking agents are all tagged or the maximum number of time steps elapses.

**Observation and Action Space**: The observations are a set of extracted features consisting of the positions, velocities, orientations and tagged status of all agents. Actions consist of accelerating forward or backward by a fixed amount, rotating left or right by a fixed offset, and tagging.

**Strategy**: Furthermore, we restricted our game such that only the defending team's policy is trained via MAPPO. While it is a straightforward extension to train both an attacker and defender policy iteratively, we opted to restrict the attacking team to sampling strategies from a fixed policy distribution to better investigate the effects of our concept policy model on performance. We sample attacker strategies from a distribution consisting of three "types" with equal probabilities, {*random*, *left*, *right*}, where the attackers execute random actions, move towards the goal by sweeping along the left side of the environment, and move towards the goal by sweeping along the right, respectively. Given a sampled team level strategy, each agent then sampled an individual policy with noise from the strategy distribution, so as to generate stochastic policies.

**Concepts**: We utilized the following concepts: {*Range*, *Strategy*, *Target*, *Orientation*, *Position*}, in which *Range* is a boolean concept indicating whether the opposing agent specified by *Target* is within tagging range, *Strategy* is a categorical concept mapping to the above team-level strategies, *Target* is a categorical concept indicating an opposing agent that should be pursued, and *Orientation* and *Position* are continuous concepts encoding the relative orientation and position of each opposing agent, respectively. The hard concept policy models are trained with the full set of concepts, while the soft models only employ a subset consisting of {*Range*, *Strategy*, *Target*}.

**Real-world Equivalent**: The real-world version of our tag game is played in a 2v2 scenario on a $6' \times 6'$ play area, with four Khepera IV [32] robots. Policies are trained in the simulation environment, then executed in the real-world environment; no additional training and no few-shot conditioning

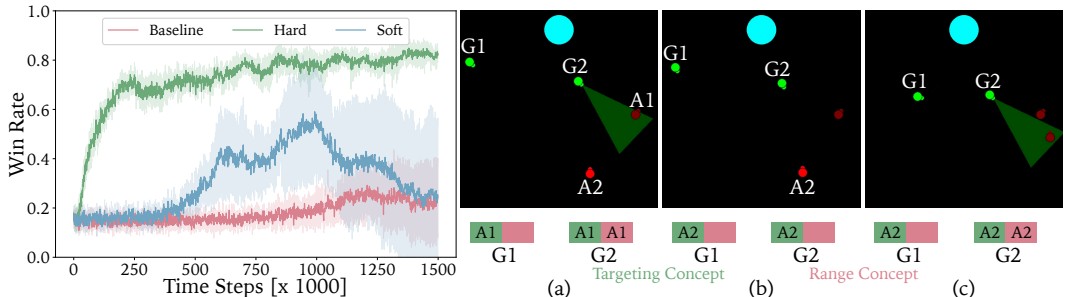

Figure 3: Left: training curves showing win rate vs iterations over 5 random training seeds for each tested model type in a 2v2 scenario. Right: a sequence of episode steps showing the concept activations for agents on the defending team (green).

is employed. The robot positions and orientations are extracted from a Vicon [33] motion capture system and converted into the model's expected observation format. The real-world version of the game exhibits significant differences in the dynamics between the simulated robots and the real-world robots – particularly in velocities, accelerations, and even control – presenting a challenging environment for sim-to-real. Further, tagged agents disappear in simulation while the real-world agents are driven out of the play area, providing a temporary obstacle.

### 5.2 Concept Policy Models and Baselines

We trained a hard and soft concept model for 10M time steps for each scenario – 2v2, 3v3, and 5v5 – along with a standard policy model without concepts. Each model consists of a series of fully connected layers, recurrent layers, and the iterative normalization layer applied over the concatenated concept and residual layers, with full details given in the supplementary material. The concept dimension $j$ for each hard model differ for each scenario due to the number of agents: $j = 13$, $j = 18$, and $j = 28$ for 2v2, 3v3, and 5v5 respectively. The concept dimensions for the soft models are $j = 9$, $j = 12$, and $j = 18$ for 2v2, 3v3, and 5v5. For the soft models, we additionally provide a residual layer with dimension $k = 23$, $k = 52$, $k = 78$ for 2v2, 3v3, and 5v5, respectively, leading to a combined bottleneck size of 32, 64, and 96. The baseline model lacks a concept layer ($j = 0$) and has a full-width residual $k = 128$. Residual layers sizes and other hyperparameters are given in the supplementary material and were chosen through extensive hyperparameter optimization.

## 6 Results

The win rates and concept accuracy errors for the defending team in both simulation and real-world are shown in Table 1. These values are computed by training two seeds with the best set of hyperparameters found during optimization, then rolling out each policy for 100 evaluation episodes in simulation, and 20 in real-world.

**Simulation**: We first observe that both concept policy model variants out-perform the baseline model in each scenario, with the hard concept model outperforming the others by a large margin. This in itself is unsurprising, given that the concepts were hand-designed so as to provide a sufficient amount of information for the policy, and the hard concept policy model heavily regularizes the learned model such that it learns this information. The decreased performance in the soft model is due to the fact that it is only trained with a subset of concepts and consequently the residual struggles to reliably encode this information on its own. As evidenced by the large win rate std shown in Table 1 on Lines 1 and 4, some seeds yield similarly strong performance to the hard models while others perform much worse – with the exception of 5v5 where they failed to learn at all. From this we can conclude that the soft concept policy models *can* offer comparable performance to hard concept models without requiring a fully-descriptive concept set, at the cost of increased training instability (see Figure 3 for a 2v2 scenario). The intervened win rate follows when an algorithmic oracle intervenes at every time step and sets the correct concept value when a concept is incorrectly predicted by the model, improving the win rate, particularly in the more complex 3v3 and 5v5 scenarios.

**Real-world**: In the real-world environment shown in Fig. 2, we can see in Table 1 on Lines 10-12 that the win rates are drastically reduced for the concept policy models, but surprisingly not for the baseline model. Qualitatively, we have observed that this is because the baseline model became

| | Setup | Model | WR | Inter. WR | Range | Strategy | Target | Orientation | Position |
|---|---|---|---|---|---|---|---|---|---|
| *Simulation* | 2v2   1 | Soft | 0.51 (0.35) | 0.55 (0.09) | 0.03 | 0.04 | 0.24 | - | - |
| | 2 | Hard | 0.83 (0.01) | 0.84 (0.01) | 0.04 | 0.07 | 0.20 | 0.10 | 0.11 |
| | 3 | Base | 0.19 (0.07) | - | - | - | - | - | - |
| | 3v3   4 | Soft | 0.55 (0.31) | 0.57 (0.27) | 0.03 | 0.10 | 0.17 | - | - |
| | 5 | Hard | 0.74 (0.01) | 0.80 (0.01) | 0.03 | 0.13 | 0.23 | 0.11 | 0.14 |
| | 6 | Base | 0.16 (0.06) | - | - | - | - | - | - |
| | 5v5   7 | Soft | 0.32 (0.01) | 0.40 (0.01) | 0.02 | 0.25 | 0.52 | - | - |
| | 8 | Hard | 0.78 (0.04) | 0.86 (0.01) | 0.03 | 0.14 | 0.13 | 0.11 | 0.21 |
| | 9 | Base | 0.31 (0.04) | - | - | - | - | - | - |
| *Real* | 2v2   10 | Soft | 0.10 | 0.00 | 0.03 | 0.53 | 0.13 | - | - |
| | 11 | Hard | 0.25 | 0.95 | 0.04 | 0.53 | 0.92 | 3.48 | 0.81 |
| | 12 | Base | 0.35 | - | - | - | - | - | - |

Table 1: The win rate (WR) and concept errors for our proposed models (Soft and Hard) and a baseline without concepts (Base). The Hard model is trained over all concepts, the Soft model over a subset, and the Base model with none. The win rate is the average standard win rate of the policy when the policy is executed over multiple seeds with the standard deviation shown in parentheses, while the intervened win rate (Inter. WR) is the average win rate when an oracle intervenes over all concepts and sets correct values. *Range*, *Strategy*, and *Target* are discrete concepts and as such the error shown is the error in accuracy score, while *Orientation* and *Position* are continuous and indicate mean squared error. *Orientation* is in radians and *Position* is a unit-less value in $[-1, 1]$.

trapped in a local minima and learned a policy which was semi-performant and independent of the actions of the opposing team, causing it to drive in circles while constantly "tagging". This behavior highlights the difficulties standard MARL algorithms have when attempting to learn meaningful feature embeddings. The other interesting result from this experiment is the gain in performance by the hard model when interventions occur, and similarly the lack of improvement when the soft model is intervened. We can draw two insights from this: the distribution shift from the simulated to the real world environment is largely contained within the feature extractor, which is compensated for by the interventions in the hard model; and that the *Orientation* and *Position* concepts are by far the most important as when we are unable to intervene on them and correct for dynamics errors as in the soft model, performance fails to improve.

**Concept Ablations**: Next, we examine ablated hard concept policy models which are only trained over a subset of concepts in the 2v2 simulated scenario. The results are shown in Table 2 (left) and further support the evidence that the *Orientation* and *Position* concepts are by far the most important with respect to the win rate. In the simulation environment, the win rate drops to 23% (Line 2) in the absence of those concepts, while in the real-world environment it drops to 27%. Note that the only difference between the hard model and the soft model with this concept set is the presence of a residual layer; when this residual is present and allowed to encode additional information the win rate is nearly doubled to 51% as in the soft model in Table 1 (Line 1).

| | Concept | | | | | Ablation | | | |
|---|---|---|---|---|---|---|---|---|---|
| | Rng. | Str. | Tgt. | Pos. | Ori. | Concept | | Interv. | |
| | | | | | | 2v2 | 3v3 | Sim | Real |
| 1 | ✓ | ✓ | ✓ | ✓ | ✓ | 83% | 74% | 84% | 95% |
| 2 | ✓ | ✓ | ✓ | | | 23% | 27% | 84% | 20% |
| 3 | ✓ | | ✓ | ✓ | ✓ | 80% | 69% | 81% | 60% |
| 4 | | | | ✓ | ✓ | 80% | 72% | 78% | 80% |

Table 2: Concept ablations when only a subset of concepts are trained in simulation, and intervention ablations in both the real and simulated 2v2 scenarios when only a subset of concepts are intervened over.

**Intervention Ablations**: We performed an ablation over the set of intervened concepts, with the results shown in Table 2 (right). We can first observe that ignoring interventions over the *Orientation* and *Position* concepts does not affect the win rate (Line 2), likely because the associated errors for those concepts are already low as shown in Table 1 (Line 2). As we intervene over fewer and fewer concepts, the win rate further drops; however, paradoxically the win rate drops to below the base win rate without any interventions at all. In the real-world we observe more pronounced effects due to the differing dynamics, where the strategy concept is more difficult to predict and impacts the behavior more significantly. We conjecture that this is because the agents are less agile in the real-world, and errors in the predicted strategy lead to less-than-optimal positioning. We note that there is an unexpected result here: intervening over every concept but *Strategy* (Line 3) yields a lower win rate than only *Orientation* and *Position* (Line 4). This appears to be a consequence of two factors: intervening over

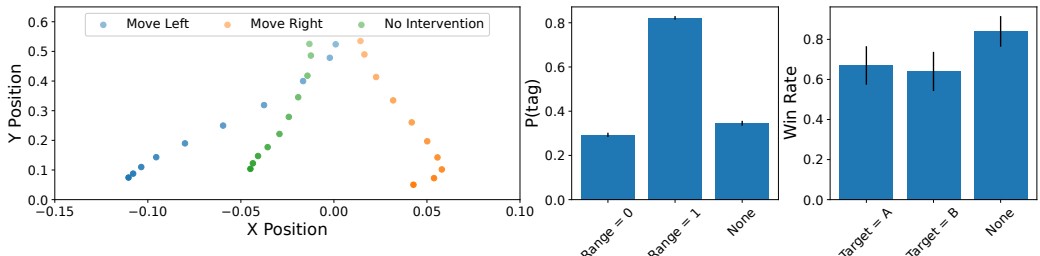

Figure 4: (Left) Intervening over *Strategy* for the 3v3 hard concept policy model in simulation. Each point is the average position of all the defenders as an episode progresses, where lighter values indicate earlier in the episode, averaged over 100 episodes. (Center) The probability of an agent performing the tag action over 1000 episodes when *Range* is predicted (None) or intervened (Range = 0/1) in a simulated 2v2 scenario. (Right) The win rate of the defending team over 1000 episodes when *Target* is predicted (None) or intervened (Target = A/B) such that both defenders share the same target in a simulated 2v2 scenario. The mean and 95% concept intervals are shown.

*Range* and *Target* but not *Strategy* increases the prediction error of *Strategy* which in turn lowers the win rate; and the policy resulting from intervening over only *Orientation* and *Position* results in more collisions with other agents during execution, making them easier to tag.

**Behavioral Effects**: Figure 4 shows the effects of concept values on concept policy model behavior. The left figure visualizes the average position of the defending team as an episode progresses when the *Strategy* concept is intervened to always take a specific value, showing that the Strategy concept is clearly correlated with the defending team's movement. Specifically, when the attacking team is predicted to move to the left or the right, the defending team moves in the corresponding direction and towards the attackers to intercept. The two figures on the right further illustrate behavioral effects resulting from concept values; the middle figure shows that the probability of performing the tag action dramatically increases when the *Range* concept is intervened and forced to 1, while the right figure shows that the win rate decreases when the agents are forced to have the same *Target* rather than predicting their own independently.

These results reveal that high-level concepts can impart meaningful behavioral effects on the downstream policy that align with the concept's interpretation; thus while the concepts do not provide full transparency into the policy's decision making process they still provide the ability for an expert to understand what high-level pieces of information are informing a policy and how these might influence behavior.

**Limitations**: In order to analyze the performance of our model for a single team of agents, we have restricted the variability in our environment and reduced the complexity of possible behaviors. In the future, we would like to evaluate asymmetric team compositions and learn a policy for the attackers. We have also only considered low-dimensional inputs in our experiments, and although we expect our approach to scale well to rich input representations such as images, since concept models have been traditionally applied in vision domains, this remains an open question. Additionally, we would like to expand the complexity of our real-world environments by incorporating additional robots.

## 7 Conclusion

In this work we have introduced *concept policy models* for Multi-Agent Reinforcement Learning which incorporate domain knowledge from an expert in the form of concepts. Concept policy models allow an expert to query the model at run-time and analyze policy behavior in terms of high-level concepts, and crucially, intervene and correct concept predictions when errors occur. We empirically show that concept policy models regularize the underlying policy, yielding improved accuracy and sample efficiency while stabilizing training, and demonstrate how oracle-based interventions can be leveraged to partially compensate for distributional shift in sim-to-real transfer scenarios. We find that such interpretability is particularly important in multi-agent settings as even small changes in agent performance can lead to large changes in team coordination. Allowing an expert to understand what high-level concepts are used by a policy and reason about how these concepts affect individual agent behavior provides insight into team dynamics, and provides a mechanism for intervening and correcting behavior when necessary.

**Acknowledgments**

This work was supported by DARPA award HR001120C0036, AFRL/AFOSR award FA9550-18-1-0251, AFOSR award FA9550-15-1-0442, ARL W911NF-19-2-0146, NSF IIS-1724222, and the National Science Foundation Graduate Research Fellowship under Grant No. 1745016.

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
