# OpenReview forum: "Concept Learning for Interpretable Multi-Agent Reinforcement Learning"
_robot-learning.org/CoRL/2022/Conference — CoRL 2022 Poster_

### Official Review · Reviewer_avMX · 2022-06-29

**Originality:** Very Good
**Technical Quality:** Very Good
**Clarity Of Presentation:** Very Good
**Impact:** 4

**Recommendation:**

Strong Accept: I recommend accepting the paper and will argue for my recommendation even if other reviewers hold a different opinion.

**Summary:**

This paper proposed to adopt the concept bottleneck network to multi-agent RL. The network architecture induces interpretability and stabilizes training, according to experimental demonstration.

**Issues:**

This is more of a discussion point, but in Eq. 2 (with notations simplified), the fact that $\pi^2$ is applied on $r(x)$ could arbitrarily amplify its effect. Is it possible to change the equation to $\pi^2(c(x)) + r(x)$, so that the contribution of the residual network is much easily quantifiable?

**Quality Of The Limitations Section:**

Limitations are addressed clearly

**Reviewer Expertise:**

4: The reviewer is confident but not absolutely certain that the evaluation is correct

**Robotics Focus:**

Sufficient demonstration on hardware

**Strengths And Weaknesses:**

Strengths:

1. This paper is genreally well written and easy to understand.

2. The motivation is strong.

3. The experimental results are strong and convincing.

Weaknesses:

1. It could help to present a diagram of the network architecture described in Eq. 2.

2. I would recommend adding parenthesis to the superscript in the policy, as $\pi^{(1)}_{\theta 1}$. Otherwise, the superscript is easy to be confused as power exponent.

3. The network architecture does have limited novelty above the conept bottleneck network, but it is refreshing to see it being adopted to MARL.

**Summary Of Recommendation:**

I recommend this paper for acceptance. In my opinion, this paper contains sufficient ideas and executions for publication.

---

> ### Author Response · Authors · 2022-08-20
> **Response to Reviewer avMX**
>
>
> We thank the reviewer for their thoughtful comments and positive feedback about our work, it is greatly appreciated. We also thank the reviewer for finding our experiments to be strong and convincing regarding the claims of our method.  While in general we agree that our method adapts concept bottleneck models to the MARL case, a significant modification that we make to this approach is the addition of a “residual layer” which was previously missing in concept bottleneck models. This relaxes a significant previous assumption of prior work: that the concept set present in the bottleneck layer must be sufficient for the downstream network to make its predictions. Intuitively, this means that a user is free to define as many or as few concepts as they like, if these concepts are insufficient for making downstream prediction tasks in the policy then an appropriately-sized residual layer will allow the concept policy model to learn its own appropriate features to use in conjunction with the defined concepts. Note that we implement a whitening operation such that the defined concepts (whatever they may be) are distinct from the learned features. To our knowledge, we are the first to use a whitening layer to enable interventions on soft bottlenecks.
>
> Additionally, we agree that a network architecture diagram is important and as such have already included one in Appendix E. We would like to include it in the main paper, however, we already have significant space constraints and so we’re not entirely sure if this will be possible.
>
> We would also like to thank the reviewer for their comments about improving notation. We absolutely agree with these observations and have updated the manuscript to reflect this. Thank you.

---

> ### Author Response · Authors · 2022-08-27
> **Followup Response to Reviewer avMX**
>
> **(1) This is more of a discussion point, but in Eq. 2 (with notations simplified), the fact that $\pi^2$ is applied on $r(x)$ could arbitrarily amplify its effect. Is it possible to change the equation to $\pi^2(c(x))+r(x)$, so that the contribution of the residual network is much easily quantifiable?**
>
> Thank you for this comment. We agree that $\pi^2$ could amplify the effect of $r(x)$, however, we do require learned policy layers to take both the residual layer and the concept layer and use them to predict the policy’s action output distribution. In our formulation, $\pi^2$ outputs the action output and so requires both the concept and residual layers as input.

---

### Official Review · Reviewer_EPqJ · 2022-07-26

**Originality:** Good
**Technical Quality:** Good
**Clarity Of Presentation:** Good
**Impact:** 3

**Recommendation:**

Weak Accept: I recommend accepting the paper, but will not argue for my recommendation if the majority of other reviewers have a different opinion.

**Summary:**

This paper adds additional supervised signals to support the policy training (PPO) in a multi-agent RL setting. In detail, they introduce classifiers and regressors to predict concrete features (concepts) like relative positions or the strategy of enemies and train these modules in a supervised way. The output actions are conditioned on these predicted features. In this way, they state the interpretability of the policy is improved. Experimental results show that this method helps improve the defending team's performance in the tag game in both simulation and real robotics.

**Issues:**

They use lots of sentences (e.g., the whole Sec.4.2) to describe the soft model. But the soft model works much worse than the hard one. In L250 they explain that this is because soft model uses fewer concepts. In this case, I think an experiment where soft model uses all concepts like the hard one is needed for a clear comparison with hard model. Otherwise, I’d consider it as an interesting but not so important variant, as the soft model’s performance is much worse, and the soft model is more complex than the hard one in both architecture and training way. In this case, the content of the soft model should be reduced in the main paper and maybe moved to the appendix.

Other small writing issues:

1. In Eq.2, I think $\pi^1_{\theta1}$ doesn’t output a distribution over the action. Therefore, it should be $x=\pi^1_{\theta1}(z_a)$ instead of $x=\pi^1_{\theta1}(u_a|z_a)$. It is actually better to use other notation to denote $\pi^1_{\theta1}$, as it is not a policy, but the first part of the policy net (more or less like a feature extractor).

2. In L261-263, I cannot see the win rates of both the concept model and baseline model from Fig.3. This sentence should be rewritten.


**Quality Of The Limitations Section:**

Additional details required

**Reviewer Expertise:**

3: The reviewer is fairly confident that the evaluation is correct

**Robotics Focus:**

Sufficient demonstration on hardware

**Strengths And Weaknesses:**

I think the research topic is important in reinforcement learning. The writing is in general easy to follow. The work has hardware experiments on real robots.

The main weakness from my perspective is, whether their method indeed improves the interpretability largely, or their method just shows that a better feature extractor helps the performance, which is well-known in deep learning. From Tab.2 we can see that the most important ‘concepts’ are relative position and orientation. Without these two concepts, the win rate drops dramatically from 83% to 23%. The other concepts only affect 3%. As the original observation already contains the absolute position and orientation, knowing the relative version doesn’t improve any ‘interpretability’ of the policy. Converting the absolute version to the relative one is more like input preprocessing. Besides, it is already well known that relative position and orientation are better than the absolute version in such a setting.
I think I agree that predicting the ‘target’ can improve the interpretability of the policy. However, according to their experiments, this concept doesn’t affect the performance so much. I think they need a better experiment setting (like more interesting concepts) to solve this issue. Besides, I think they can design some experiments to show the improved interpretability of the policy directly. For example, they can show that when they intervene in the ‘target’ concept, the defender agent will change their tagging target in the environment.


**Summary Of Recommendation:**

I feel what they have done is more like using supervision signals to train a better feature extractor that helps performance instead of introducing something which clearly improves the interpretability of the policy. They should have some experiments to demonstrate the improved interpretability directly. In addition, in their experiments, the concepts that actually matter are relative position and orientation. It is better to have experiments with more interesting concepts.

**After Rebuttal**
I appreciate the authors' efforts on the new experiments in Appendix D that show the interpretability directly. I have increased my score to weak accept. I'll suggest authors include more experimental results that demonstrate the improved interpretability in the main paper.

---

> ### Author Response · Authors · 2022-08-20
> **Response to Reviewer EPqJ: Part 1**
>
> We thank the reviewer for their valuable and detailed feedback, particularly for acknowledging the relevance of our contribution to the general domain of reinforcement learning and the real-world experiments.
>
> We would like to begin by addressing some of the comments regarding interpretability.
>
> **(1) The main weakness from my perspective is, whether their method indeed improves the interpretability largely, or their method just shows that a better feature extractor helps the performance, which is well-known in deep learning. From Tab.2 we can see that the most important ‘concepts’ are relative position and orientation. Without these two concepts, the win rate drops dramatically from 83% to 23%. The other concepts only affect 3%. As the original observation already contains the absolute position and orientation, knowing the relative version doesn’t improve any ‘interpretability’ of the policy. Converting the absolute version to the relative one is more like input preprocessing. Besides, it is already well known that relative position and orientation are better than the absolute version in such a setting. I think I agree that predicting the ‘target’ can improve the interpretability of the policy. However, according to their experiments, this concept doesn’t affect the performance so much.**
>
> If we understand correctly, the arguments being made here are: a) some concepts are subjectively less interpretable than others, b) some concepts are significantly more useful to the downstream policy than others, and c) if the useful concepts are not subjectively more interpretable than others, then this reduces to feature extraction.
>
> Regarding (a), we agree that not all concepts provide an equal level of interpretability; some concepts are more interpretable in our experiments, e.g., strategy, and some are less interpretable, e.g., relative orientation. However, we do argue that taken as a whole, being able to observe which concepts the policy is using to make a decision inherently makes it more interpretable than having access to none (and we note that concept-based models have long been explored in explainable AI in traditional settings such as classification [2][3][4]). One of the strengths of our approach is that the domain expert is free to define whatever concepts they wish; we impose no constraints other than that they must be continuous or discrete labels. If no meaningful concepts have been defined then that is an issue with the domain expert, and in our case we argue that we have defined several meaningful concepts: strategy, target, and range at a minimum.
>
> For (b), it is also a given that some concepts will be more useful than others for the downstream policy. However, we note that the usefulness of concepts depends heavily on the environment dynamics. While the reviewer is correct that the range, strategy, and target concepts only seem to contribute to 3% of the win rate in the simulated 2v2 scenario in Table 2 (upper left), we can see that in the real-world 2v2 scenario (lower right) that failing to intervene over the strategy concept causes a 45% drop in win rate (95% to 50%). This means that the strategy plays a very important role with real-world dynamics. We conjecture that this is because the agents are far more maneuverable in simulation than they are in the real-world, so any positional disadvantage caused by mis-predicting the opposing team’s strategy in simulation can be made up for later, whereas in the real-world this is not the case.
>
> This leads us to (c), where we argue that this is not simply feature extraction. For one, the strategy concept in the real-world scenarios is extremely important (affecting win rate by up to 45%), and is more complex than just an absolute position to relative position mapping. Furthermore, with our approach the domain expert doesn’t need to worry about defining the set of concepts that are both highly interpretable and fully descriptive, as they would need to in a simple feature extraction case. They can define as few or as many concepts as desired, and set the residual layer to a non-zero size if they are not fully descriptive. To re-iterate some of our comments we made in our response to Reviewer p3Pf which contrasts a simple feature extraction method to our approach.
>
> (Continued in part 2)

---

> > ### Comment · Reviewer_EPqJ · 2022-08-25
> > **Reply**
> >
> > Thank you for your answer to my questions.
> >
> > ** In Table 2 (upper left), we can see that in the real-world 2v2 scenario (lower right) that failing to intervene over the strategy concept causes a 45% drop in win rate (95% to 50%). This means that the strategy plays a very important role with real-world dynamics.**
> >
> > I notice that in this experiment when you further remove the concept Range and Target and keep the Ori. and Pos only (the OP model in Tab. 2), the performance increases from 50% to 80%. To me, the OP model shows that the three concepts Strategy, Range, and Target only contribute about 15% of the performance and are not as important as shown in the RTOP model. Can you elaborate more on this?
> >
> > **The result is that some seeds perform very well and some seeds very poorly and so the average performance across multiple seeds reported in Table 1 reflects this. We will add new values to Table 1 indicating the performance of the best seed**
> >
> > I will actually have concerns if you tune the random seeds and cherry-pick the best one. I think a better way is to add the standard deviation to Tab.1 to show that the soft model has the potential to achieve good results, with the cost of training stability. Besides, I notice that in Fig.2 where the error bars are plotted, even the upper boundary of the soft model curve is clearly worse than the hard one.

---

> > > ### Author Response · Authors · 2022-08-27
> > > **Response to Reviewer EPqJ**
> > >
> > > **(1) I notice that in this experiment when you further remove the concept Range and Target and keep the Ori. and Pos only (the OP model in Tab. 2), the performance increases from 50% to 80%. To me, the OP model shows that the three concepts Strategy, Range, and Target only contribute about 15% of the performance and are not as important as shown in the RTOP model. Can you elaborate more on this?**
> > >
> > > This is a great observation, and we have performed analysis to try and investigate what is happening with this result. We conjecture that this appears to be a consequence of two factors: 1) the RTOP model has a higher prediction error for Strategy than the OP model, and 2) we have observed from our recorded experiment videos that the OP policy results in more collisions between agents and other agents (or walls).
> > >
> > > Regarding 1, it is clear that in the real-world the strategy concept is an important concept for win rate. We suspect this is due to the reduced agility of the agents in the real-world and also the higher prediction error for this concept compared to simulation. The strategy concept correlates with policy behavior (see Fig. 4 (left)), and so a mis-predicted strategy can cause the agents to become out of position which makes it difficult to stop the attackers. When we measured the strategy concept error for RTOP, we find that it is ~10% higher than for the OP model. So interestingly, intervening over range and target affects the policy behavior in such a way that strategy becomes harder to predict.
> > >
> > > Regarding 2, our robots have a collision avoidance algorithm enabled when they come too close to each other (on the order of centimeters) which will override policy actions. Going back and observing the experiment videos, we have observed that the RTOP models result in collision avoidance activating 2 times while for the OP models it activates 5 times. The collision avoidance algorithm tends to make the attacking agents easier to tag, as they slow down and turn, and in some cases stop entirely as they become blocked in place by other agents. This can result in the OP policy getting some tags more easily due to near collisions.
> > >
> > > **(2) I will actually have concerns if you tune the random seeds and cherry-pick the best one. I think a better way is to add the standard deviation to Tab.1 to show that the soft model has the potential to achieve good results, with the cost of training stability. Besides, I notice that in Fig.2 where the error bars are plotted, even the upper boundary of the soft model curve is clearly worse than the hard one.**
> > >
> > > Thank you for this feedback. We have added the standard deviation to the seeds in Table 1 for Win Rate and Intervened Win Rate for simulation (we only tested 1 seed in the real-world). In addition, we note here that the underlying data shows that some seeds yield similar performance to the hard concept policy models – 76% win rate vs 84% win rate for 2v2 and 77% vs 74% for 3v3 for soft and hard models respectively – while other seeds perform worse due to instability during training. The soft concept policy models struggle to learn an effective policy in the 5v5 scenario across all seeds though.
> > >
> > > Regarding Figure 2 (now Figure 3), the upper boundary of the soft model matches the performance of the hard model at the ~1000 mark (1 million time steps). Performance is of course unstable afterwards, but if we were to deploy policy models in the real world it is common to take the best performing checkpoint during training.

---

> ### Author Response · Authors · 2022-08-20
> **Response to Reviewer EPqJ: Part 2**
>
> Let’s consider a common alternative approach: directly using the concepts as inputs to the policy network, including any conditional inputs/commands as in [1]. This leads to the following differences.
>
> 1. If concepts are provided as the sole inputs to a policy, then the concept set must sufficiently describe the state for any downstream tasks, i.e., the policy. In our approach, we provide an optional residual layer, such that the domain expert can provide as many or as few concepts as they like, and if they are not descriptive enough for the downstream task then they may set a non-zero size for the residual layer such that it can independently capture the additional necessary information. In the event the concepts are fully descriptive, a hard concept policy model is sufficient, while in all other cases a soft concept policy model should be utilized.
>
> 2. If an additional observation is provided in addition to the concepts to overcome the descriptive issue in (1), then there is nothing preventing features learned from the observation from encoding the same information as the concepts. In such a case, interventions are unlikely to be effective since the policy may base its decision only partly on the concepts (if at all). In our approach, because we whiten the residual and concept layers we force the residual to encode non-concept information.
>
> 3. Concepts must now be provided at every time step, including during test time. In the case of [1], this means there’s no way for the agent to infer its own goal (and make it available for a domain expert to double check) unless the goal is also inferred by the feature extractor, but then you have the problem of (1) above where the goal may be ignored. In our approach, concepts must only be provided during training, yet the predicted values can be observed and intervened over during testing.
>
> 4. If concepts are required as inputs to the policy, this means they need to be calculated up-front as part of the observation before a policy step is taken in the environment. This could pose complications in an RL setup when policies are rolled out since some concepts may not be obvious until future time steps, e.g., which agent is being targeted in the very first time steps of an episode. So a), the feature extractor would need to be trained ahead of time or b) it would need to be trained simultaneously with the policy network, where (b) reduces to a concept policy model (albeit without whitening to address the problem described in (1)). In our approach, concepts do not need to be provided as part of the observation and the upstream feature encoder layers are trained simultaneously with the policy layers.
>
> Our approach is novel in that it addresses all of these issues: a) the concept set does not need to fully describe the state for downstream tasks, b) residual information does not re-encode the concepts, c) concepts only need to be provided at training time, and at test time the predicted concepts can be observed and – most importantly – intervened, and d) concepts are trained simultaneously with the policy and labels can be computed post-hoc at the end of an episode. To our knowledge, we are the first to use a whitening layer to enable interventions over concepts in reinforcement learning policies.

---

> ### Author Response · Authors · 2022-08-20
> **Response to Reviewer EPqJ: Part 3**
>
> **(2) However, according to their experiments, this concept doesn’t affect the performance so much. I think they need a better experiment setting (like more interesting concepts) to solve this issue. Besides, I think they can design some experiments to show the improved interpretability of the policy directly. For example, they can show that when they intervene in the ‘target’ concept, the defender agent will change their tagging target in the environment.**
>
> In addition to the strategy concept in the real-world scenarios as described above, we currently have experiments in Appendix D which show the effect of intervening over certain concepts on the final behavior of the policy. In particular, we show that when we intervene and always set the Range concept to True, that the agent becomes far more likely to perform the “tag” action than when this is not the case (~35% to ~85%). Additionally, when we intervene and set the strategy concept to left or right, we can observe a meaningful difference in the average position of the defending agents: when set to left the defending agents move left to intercept, when right they move to the right.
>
> We think these experiments provide evidence that intervening and setting concept values indeed affects final policy behavior, but we also propose to add a new experiment where we intervene and override the target concept to a random target or alternatively to all focus on a single target. We observe a meaningful difference in behavior as a result, and will include the results in an updated manuscript. Would this be of interest to the reviewer?
>
> **(3) They use lots of sentences (e.g., the whole Sec.4.2) to describe the soft model. But the soft model works much worse than the hard one. In L250 they explain that this is because soft model uses fewer concepts. In this case, I think an experiment where soft model uses all concepts like the hard one is needed for a clear comparison with hard model. Otherwise, I’d consider it as an interesting but not so important variant, as the soft model’s performance is much worse, and the soft model is more complex than the hard one in both architecture and training way. In this case, the content of the soft model should be reduced in the main paper and maybe moved to the appendix.**
>
> One challenge with the soft concept policy models are that the whitening operation leads to training instability when compared to the hard concept policy models. The result is that some seeds perform very well and some seeds very poorly and so the average performance across multiple seeds reported in Table 1 reflect this. We will add new values to Table 1 indicating the performance of the best seed, and show that the soft concept policy models are able to achieve better performance in practice.
>
> **(4) Notational feedback**
>
> We greatly appreciate the comments on improving notational clarity and agree with them. We will make these changes and post an updated manuscript version. Thank you.
>
> References:
>
> [1] Codevilla, F., Müller, M., López, A., Koltun, V. and Dosovitskiy, A., 2018, May. End-to-end driving via conditional imitation learning. In 2018 IEEE international conference on robotics and automation (ICRA)
>
> [2] Koh, P.W., Nguyen, T., Tang, Y.S., Mussmann, S., Pierson, E., Kim, B. and Liang, P., 2020, November. Concept bottleneck models. In International Conference on Machine Learning
>
> [3] Chen, Z., Bei, Y. and Rudin, C., 2020. Concept whitening for interpretable image recognition. Nature Machine Intelligence
>
> [4] Kim, B., Wattenberg, M., Gilmer, J., Cai, C., Wexler, J. and Viegas, F., 2018, July. Interpretability beyond feature attribution: Quantitative testing with concept activation vectors (tcav). In International conference on machine learning

---

### Official Review · Reviewer_dgWp · 2022-08-03

**Originality:** Good
**Technical Quality:** Good
**Clarity Of Presentation:** Fair
**Impact:** 2

**Recommendation:**

Weak Accept: I recommend accepting the paper, but will not argue for my recommendation if the majority of other reviewers have a different opinion.

**Summary:**

To improve the interpretability of the RL policy model, This paper enforces the policy model to encode the "concept" in the intermediate layers of the model. The concepts are categorical or continuous values relevant to the task. For example, in competitive games, the opponent's strategy (offensive/defensive) and range (0~100) can be the concepts. The policy model consists of two major modules. The first model predicts the concepts based on observations. Domain experts can correct the predicted concepts. The second module transforms the concept's value into actions. The advantage is that human experts can intervene in the concept predictions and correct them. The experimental results show performance improvements.

**Issues:**



**Quality Of The Limitations Section:**

Limitations are addressed clearly

**Reviewer Expertise:**

2: The reviewer is willing to defend the evaluation, but it is quite likely that the reviewer did not understand central parts of the paper

**Robotics Focus:**

Sufficient demonstration on hardware

**Strengths And Weaknesses:**

Pros:
- The idea of using concept learning on sim-to-real and interpretability is interesting.

Cons:
- Section 6 is very hard to parse -- especially real-world results. I suggest making those results in the real world as figures or tables instead of describing them in a dense paragraph.
- Lack of experiments. The experiments are limited -- only in 1 task.

**Summary Of Recommendation:**

I recommend weak acceptance since the author successfully validates the claim described in the paper and there is no obvious technical flaws.

---

> ### Author Response · Authors · 2022-08-20
> **Response to Reviewer dgWp**
>
> We would like to thank the reviewer for their thoughtful comments and recognition of the strength of our
> explainable concept based learning approach.
>
> To address specific issues:
>
> **(1) Section 6 is very hard to parse -- especially real-world results. I suggest making those results in the real world as figures or tables instead of describing them in a dense paragraph.**
>
> We appreciate and agree with the feedback regarding Sec. 6, and will be providing an updated manuscript version in the coming days in which we provide more figures and improve the clarity of our results analysis.
>
> **(2) Lack of experiments. The experiments are limited -- only in 1 task.**
>
> While we think our scenario is an interesting and challenging one, we acknowledge that additional experimental tasks would help in analyzing the performance of our approach. We therefore plan to run additional simulated experiments using the Petting Zoo multi-agent particle environments, specifically the predator-prey environment, and provide the results in an updated manuscript. Would this be satisfactory?

---

> ### Author Response · Authors · 2022-08-27
> **Followup Response to Reviewer dgWp**
>
> Thank you again for your helpful feedback. We are following up with specific comments after revising the manuscript.
>
> **(1) Section 6 is very hard to parse -- especially real-world results. I suggest making those results in the real world as figures or tables instead of describing them in a dense paragraph.**
>
> We have introduced changes to the manuscript to help improve the readability. In particular we have adjusted the layout of Section 6 and added line numbers to Tables 1 and 2 which are referred to in the text.
>
> **(2) Lack of experiments. The experiments are limited -- only in 1 task.**
>
> We are currently running additional experiments on the Predator-Prey environment in MAPE. However, integrating these environments into our code base has proven time consuming and so we do not have the results ready at this time for the current version of our paper. We expect these results to be available sometime next week and will integrate them into the paper.

---

### Official Review · Reviewer_p3Pf · 2022-08-10

**Originality:** Fair
**Technical Quality:** Good
**Clarity Of Presentation:** Fair
**Impact:** 3

**Recommendation:**

Weak Reject: I recommend rejecting the paper, but will not argue for my recommendation if the majority of other reviewers have a different opinion.

**Summary:**

This work investigates the interpretability of multi-agent RL and introduces a method for incorporating interpretable concepts from domain expert.

The concepts can be considered as hidden states of current environment.

The proposed Concept Policy Model decomposes the policy network into several parts:

1. A feature extractor,
2. A residual layer that produces "residual vector" that has no information related to concept,
3. A concept layer that produces "concept vector",
4. Policy layer produces action distribution.

For the concept layer, supervised learning is conducted to fit the concept vector to ground-truth concepts provided by environment.

For the residual vector, the ZCA whitening is used to manipulate the activation matrix and let the output does not contain information about the concepts.

The concept vector output by the concept layer serves as part of the input to the final policy layer. This means that (1) we can query and manipulate the concept during execution.

**Issues:**


### Method

1. For the residual vector, the ZCA whitening is used to manipulate the activation matrix and let the output of residual vector does not contain information about the concepts. I am not fully understand what is happening during this process. Could you explain more here?


2. Line 231: "The iterative normalization layer is applied over the concatenated concept and residual layers." IIUC, the concept layer should not be "whitened"?


3. Is it possible that the concepts you selected as described in "Concepts" (Line 212) are very strong and informative observation? That is to say, consider a extreme case where your concept layer is perfectly trained, then it should always output ground-truth concepts as described in Line 212. Those information is strong enough, along side with the information came from residual layer that is extracted from the observation, to learn a powerful agent. To justify this, I will expect a new experiment that use directly the ground-truth concepts as input (replacing the output of concept layer) to the policy layer.

4. How you overwrite the activation of concept layer? (Sec 4.3)


5. Other issues on method is in "Summary Of Recommendation".






### Writing

1. Line 59, 112, 207: Wrong latex code for left quotation mark
2. Table 1 is too far from Result section.
3. I would expect to see a figure to illustrate the network architecture of Concept Policy Models in Sec 4.1
4. Line 310 - Line 314: A long and non-informative sentence. I don't think it provide any interesting conclusion to reader.



**Quality Of The Limitations Section:**

Additional details required

**Reviewer Expertise:**

4: The reviewer is confident but not absolutely certain that the evaluation is correct

**Robotics Focus:**

Sufficient demonstration on hardware

**Strengths And Weaknesses:**


### Strengths

* The interpretability of MARL agents is a valuable research topics. I am happy to see work pursing this.
* The idea that "human can intervene when the agent understands the environment in a wrong way" is inspiring. This reminds me works on intervention-based imitation learning and model-based learning. The concept layers in this work is just like the dynamic models (both are trained with environment-provided ground truth information).


### Weaknesses

* The clarity is poor. As an example, in Sec 4.2, ZCA whitening is conducted on the activation values of residual values. The motivation here is to let "residual layer not to encode information related to concepts" (line 152). What is the connection between "let it not encode something" and "use ZCA whitening"? Also, what is the $\mu_x$ in Equation 4?
* The motivation and whole story are disconnected from the method and experiment that really implemented. See "Summary Of Recommendation".
* The experiment is oversimplified (See "Issues, Method, Q3") and the "concept" is a concept that poorly defined.

**Summary Of Recommendation:**


Generally, I think the method proposed in this work is far less novel and impacting than it is claimed in this work.


I admit that the problem trying to be addressed is important, the interpretability of MARL algorithms.
I would expect to see the authors take the unique feature of the problem, multi-agent, into the consideration and propose a method that can infer the causality between agents' behaviors.


However, this work proposes to decompose the policy network and use extra supervision signal to force the concept layer to predict ground truth concept. Conditioning the policy network with "high-level concept" is not a novel idea and is widely used before, such as [1].
It turn out to be a method that adding supervision loss to fit human-labelled concepts and there is no any special design for MARL setting.

[1] End-to-end driving via conditional imitation learning

The mismatch between what it claims and what it does is the major reason for my recommendation.

1. In conclusion, the authors claim "Concept Policy Models incorporate domain knowledge from an expert" (Line 310). Why don't you just use those concept as input directly to the policy network? (See Issues - Method - Q3) Also, the proposed method is not "a general framework" (Line 311) since it requires hand-coded "concepts" from the environment and the "domain of concepts" is designed by human (I can accept this, it is inevitable that we need some prior knowledge. However, this makes the proposed method unqualified to be a "general framework").
2. In abstract, the authors claim "This allows a human operator to both obtain decision making rationale and intervene" (Line 8). However, I can't find the description on how human expert intervene and improve the wining rate. I am doubting whether a human expert can really figure out the logic of agents' behavior by watching per-step predicted concepts (which, basically are a set of informative values of current environment, like locations of nearby enemies).
3. As I discussed above, I can't find any particular design to make the proposed method more suitable in MARL setting.

In conclusion, I think current submission is below the bar of acceptance.

---

> ### Author Response · Authors · 2022-08-20
> **Response to Reviewer p3Pf: Part 1**
>
> We would like to thank the reviewer for their thorough and thoughtful comments, particularly the positive feedback on our intervenable concept policies. The comments regarding presentation and clarity are especially helpful and will help us produce a more effective paper, and for that we are thankful.
>
> We would first like to re-emphasize our novelty claims, and particularly how our approach differs from prior work (including [1] which the reviewer kindly pointed out). Intuitively, our proposed concept policy model takes a standard end-to-end policy network architecture and introduces an additional layer into the middle of it, essentially separating the network into two parts: the upstream part (feature encoding layers) and downstream part (policy layers). This additional layer itself consists of two parallel pieces: a residual layer and a concept layer. The concept layer predicts semantically meaningful concepts which are defined by a domain expert and trained through the addition of an auxiliary loss. The residual layer remains unconstrained (no auxiliary loss) and exists to capture any non-concept information necessary for the downstream policy layers. We enforce the constraint that the residual layer encodes non-concept information through the use of a whitening operation (iterative normalization).
>
> Now let’s consider a common alternative approach: directly using the concepts as inputs to the policy network, including any conditional inputs/commands as in [1]. This leads to the following differences.
>
> 1. If concepts are provided as the sole inputs to a policy, then the concept set must sufficiently describe the state for any downstream tasks, i.e., the policy. In our approach, we provide an optional residual layer, such that the domain expert can provide as many or as few concepts as they like, and if they are not descriptive enough for the downstream task then they may set a non-zero size for the residual layer such that it can independently capture the additional necessary information. In the event the concepts are fully descriptive, a hard concept policy model is sufficient, while in all other cases a soft concept policy model should be utilized.
>
> 2. If an additional observation is provided in addition to the concepts to overcome the descriptive issue in (1), then there is nothing preventing features learned from the observation from encoding the same information as the concepts. In such a case, interventions are unlikely to be effective since the policy may base its decision only partly on the concepts (if at all). In our approach, because we whiten the residual and concept layers we force the residual to encode non-concept information.
> 3. Concepts must now be provided at every time step, including during test time. In the case of [1], this means there’s no way for the agent to infer its own goal (and make it available for a domain expert to double check) unless the goal is also inferred by the feature extractor, but then you have the problem of (1) above where the goal may be ignored. In our approach, concepts must only be provided during training, yet the predicted values can be observed and intervened over during testing.
> 4. If concepts are required as inputs to the policy, this means they need to be calculated up-front as part of the observation before a policy step is taken in the environment. This could pose complications in an RL setup when policies are rolled out since some concepts may not be obvious until future time steps, e.g., which agent is being targeted in the very first time steps of an episode. So a), the feature extractor would need to be trained ahead of time or b) it would need to be trained simultaneously with the policy network, where (b) reduces to a concept policy model (albeit without whitening to address the problem described in (1)). In our approach, concepts do not need to be provided as part of the observation and the upstream feature encoder layers are trained simultaneously with the policy layers.
>
> Our approach is novel in that it addresses all of these issues: a) the concept set does not need to fully describe the state for downstream tasks, b) residual information does not re-encode the concepts, c) concepts only need to be provided at training time, and at test time the predicted concepts can be observed and – most importantly – intervened, and d) concepts are trained simultaneously with the policy and labels can be computed post-hoc at the end of an episode. To our knowledge, we are the first to use a whitening layer to enable interventions over concepts in reinforcement learning policies.

---

> ### Author Response · Authors · 2022-08-20
> **Response to Reviewer p3Pf: Part 2**
>
> We will now address specific issues.
>
> **(1) For the residual vector, the ZCA whitening is used to manipulate the activation matrix and let the output of residual vector does not contain information about the concepts. I am not fully understand what is happening during this process. Could you explain more here?**
>
> We concatenate the residual layer and concept layer activations over a mini-batch and perform iterative normalization [2]. This process iteratively decorrelates the activations by applying Newton’s method to approximate the whitening matrix, where T is the number of iterations. The end result is that for a given mini-batch, the activation vectors are whitened such that the output of this combined layer has an approximately isometric diagonal covariance, where the closeness of this approximation is related to the number of iterations. Intuitively, this forces each node to encode a different piece of information than each other node, thus preventing the residual nodes from re-encoding the information present in the concept nodes. We will add a technical discussion on this topic to the appendix.
>
> **(2) Line 231: "The iterative normalization layer is applied over the concatenated concept and residual layers." IIUC, the concept layer should not be "whitened"?**
>
> This is a great observation, and indeed intuitively it seems odd to decorrelate the concept layer activations with each other. However, in practice this works reasonably well [3] and simply decorrelating all activations allows for the efficient computation of the whitening operation [2], although it can cause instability if the defined concepts are too closely correlated. We note that this is actually the cause of some of the instability with our soft concept policy models – some seeds perform very well while other seeds do not – which brings down the overall average performance of our soft bottleneck models in Table 1. We will add additional values to Table 1 showing the best seed performance to empirically show this is the case.
>
> In the future, we plan to investigate methods which only perform a one-way decorrelation, e.g., whiten the residual with respect to the concept.
>
> **(3) Is it possible that the concepts you selected as described in "Concepts" (Line 212) are very strong and informative observation? That is to say, consider a extreme case where your concept layer is perfectly trained, then it should always output ground-truth concepts as described in Line 212. Those information is strong enough, along side with the information came from residual layer that is extracted from the observation, to learn a powerful agent. To justify this, I will expect a new experiment that use directly the ground-truth concepts as input (replacing the output of concept layer) to the policy layer.**
>
> Thank you for pointing this out. Our intervention experiments – see the “Intervened WR” column in Table 1 – assume the domain expert intervenes at every time step in which they should and always provides the correct ground-truth concept. We use an algorithmic oracle as a proxy for a domain expert in these experiments and we will add a statement indicating this is the case to the manuscript. With this assumption, this Intervened WR is equivalent to directly using ground-truth concepts. The intervention ablation experiment in the bottom of Table 2 modifies this experiment such that only specific concepts are intervened over (although again in every time step and with correct concepts).
>
> **(4) How you overwrite the activation of concept layer? (Sec 4.3)**
>
> In the case of a classification concept, that concept is represented as a probability distribution. Overwriting the concept would therefore be updating the distribution.  For example, for a four choice concept where the second concept is true, we'd replace the distribution with [0,1,0,0]. For updates to a regression concept, we would update the relevant neurons with their corresponding correct values.

---

> ### Author Response · Authors · 2022-08-20
> **Response to Reviewer p3Pf: Part 3**
>
> **(5) In abstract, the authors claim "This allows a human operator to both obtain decision making rationale and intervene" (Line 8). However, I can't find the description on how human expert intervene and improve the wining rate. I am doubting whether a human expert can really figure out the logic of agents' behavior by watching per-step predicted concepts (which, basically are a set of informative values of current environment, like locations of nearby enemies).**
>
> For the first part of this comment, we agree that this wasn’t properly discussed in the paper. As discussed in our response to Issue 3 above, we use an oracle as a proxy for a domain expert and assume that it intervenes every time step in which a concept is incorrectly predicted, and provides the correct ground truth concept. We will update the manuscript to make this clearer.
>
> For the second part, we don’t claim that a human expert can determine the logic of an agent’s behavior by watching the predicted concepts. Rather, it is that if the concepts are correct, then the agent should yield behavior as observed during training. In other words, correct concepts should yield expected behavior where the expected behavior is already known. This is only strictly true for hard concept policy models, as soft concept policy models may still predict incorrect features in the residual, but it does provide a mechanism for producing expected behavior as long as a domain expert is able to identify incorrect concepts and correct them (see Table 1 and Table 2). We empirically show that correcting these concepts in a sim-to-real scenario dramatically boosts policy performance.
>
> **(6) As I discussed above, I can't find any particular design to make the proposed method more suitable in MARL setting.**
>
> This is a fair point, we do not make any explicit assumptions or modifications to our approach to support the MARL setting. However, we see this as a strength: our approach works well even in MARL scenarios without requiring any additional assumptions or mechanisms. We would also like to note that interpretability is especially important in MARL due to the increased complexity of the environment dynamics, particularly dynamic agent-agent interactions such as the opposing team’s strategy. It is for this reason that we focused on the multi-agent case.
>
> **(7) I would expect to see a figure to illustrate the network architecture of Concept Policy Models in Sec 4.1**
>
> We currently have a network architecture figure in Appendix E. As we make changes we will consider moving this figure to the main manuscript, however, it may not be possible due to space constraints.
>
> **(8) Writing feedback**
>
> Thank you very much for your comments and suggestions regarding writing clarity. We will make these changes in an updated manuscript version.
>
> References:
>
> [1] Codevilla, F., Müller, M., López, A., Koltun, V. and Dosovitskiy, A., 2018, May. End-to-end driving via conditional imitation learning. In 2018 IEEE international conference on robotics and automation (ICRA)
>
> [2] Huang, L., Zhou, Y., Zhu, F., Liu, L. and Shao, L., 2019. Iterative normalization: Beyond standardization towards efficient whitening. In Proceedings of the IEEE/CVF Conference on Computer Vision and Pattern Recognition
>
> [3] Chen, Z., Bei, Y. and Rudin, C., 2020. Concept whitening for interpretable image recognition. Nature Machine Intelligence

---

> ### Comment · Reviewer_p3Pf · 2022-08-22
> **Thanks for responses**
>
> Thanks for detailed responses. My concerns are not well addressed. And the overclaim still exists. I will maintain my score.
>
> ---
>
> My question: The proposed method is not "a general framework" since it requires hand-coded "concepts" from the environment and the "domain of concepts" is designed by human.
>
> The authors do not comment on this. I think the assumption on the ability of human expert is too strong. Currently the proxy expert intervene and provide correct environmental information (so called concepts) to the system in real-time. The claim need to be toned back and narrow down. And I am still worrying about the applicability the method to real human subjects.
>
> ---
>
> My question: In abstract, the authors claim "This allows a human operator to both obtain decision making rationale and intervene" (Line 8). I can't find the description on how human expert intervene and improve the wining rate.
>
> The authors response to this issue by saying "we don’t claim that a human expert can determine the logic of an agent’s behavior by watching the predicted concepts. Rather, it is that if the concepts are correct, then the agent should yield behavior as observed during training."
>
> I think the ability for human expert to discriminate if the concept is correct is not equivalent to the ability to "obtain a decision making rationale" (Line 8). And using a "proxy of human expert" to discriminate concept is far less surprising and novel than the claim of "human can reason and intervene".
>
> ---
>
> My question: I can't find any particular design to make the proposed method more suitable in MARL setting.
>
> The authors response "we do not make any explicit assumptions or modifications to our approach to support the MARL setting. However, we see this as a strength: our approach works well even in MARL scenarios without requiring any additional assumptions or mechanisms."
>
> I don't like this excuse and it is not convincing to me.

---

> > ### Author Response · Authors · 2022-08-27
> > **Response to Reviewer p3Pf: Part 1**
> >
> > Thank you again for the detailed and helpful comments.
> >
> > **(1) The proposed method is not "a general framework" since it requires hand-coded "concepts" from the environment and the "domain of concepts" is designed by human.**
> >
> > **The authors do not comment on this. I think the assumption on the ability of human expert is too strong. Currently the proxy expert intervene and provide correct environmental information (so called concepts) to the system in real-time. The claim need to be toned back and narrow down. And I am still worrying about the applicability the method to real human subjects.**
> >
> > We have removed references to our proposed method as a “general framework” given the requirement for domain expert-specified prior knowledge.
> >
> > We have also reworded text that implies a human expert is capable of intervening and correcting concepts in real-time, instead referring to this as a “domain expert” and in particular, an “oracle”. We recognize that a human expert may be unable to match the intervention performance demonstrated by the algorithmic oracle in our results (where we assume perfect intervention accuracy at every time step), and so use the “domain expert” as a motivating goal but only claim intervention performance with an “oracle”.
> >
> > We agree that examining intervention performance with actual human participants is a valuable next step. We conjecture that final policy performance would fall somewhere between the “no intervention” and “perfect intervention” case demonstrated here, but this would need to be examined in detail with a thorough human participant study. We leave this to future work.
> >
> > **(2) In abstract, the authors claim "This allows a human operator to both obtain decision making rationale and intervene" (Line 8). I can't find the description on how human expert intervene and improve the wining rate.**
> >
> > **The authors response to this issue by saying "we don’t claim that a human expert can determine the logic of an agent’s behavior by watching the predicted concepts. Rather, it is that if the concepts are correct, then the agent should yield behavior as observed during training."**
> >
> > **I think the ability for human expert to discriminate if the concept is correct is not equivalent to the ability to "obtain a decision making rationale" (Line 8). And using a "proxy of human expert" to discriminate concept is far less surprising and novel than the claim of "human can reason and intervene".**
> >
> > This is a great comment and we agree with it. In response, we have scaled back our claims in the manuscript. We have adjusted our claims such that we no longer claim to directly produce a “decision making rationale”. Instead, we claim that our concept policy models allow a domain expert to reason about a policy in terms of these high-level concepts, as it is natural for humans to reason in high-level abstract terms rather than the low-level nonlinear feature activations typical of end-to-end models.
> >
> > To support our above adjusted claim, we have run new experiments and added the results in Figure 4 and the “Behavioral Effects” sub-section. From the paper:
> >
> > Figure 4 shows the effects of concept values on concept policy model behavior. The left figure visualizes the average position of the defending team as an episode progresses when the Strategy concept is intervened to always take a specific value, showing that the Strategy concept is clearly correlated with the defending team's movement. Specifically, when the attacking team is predicted to move to the left or the right, the defending team moves in the corresponding direction and towards the attackers to intercept. The two bar figures further illustrate behavior effects from concept values; the middle figure shows the probability of a defending agent performing the tag action when the Range concept is intervened and forced to take a value of either 0 or 1, while the right figure shows the resulting win rate when the Target concept is intervened such that both agents are forced to target the same attacking agent rather than predicting their own target independently. These results reveal that high-level concepts can impart meaningful behavioral effects on the downstream policy that align with the concept's interpretation; thus while the concepts do not provide full transparency into the policy's decision making process they still provide the ability for an expert to understand what high-level pieces of information are informing a policy and how these might influence behavior.

---

> > ### Author Response · Authors · 2022-08-27
> > **Response to Reviewer p3Pf: Part 2**
> >
> >
> > **(3) I can't find any particular design to make the proposed method more suitable in MARL setting.**
> >
> > **The authors response "we do not make any explicit assumptions or modifications to our approach to support the MARL setting. However, we see this as a strength: our approach works well even in MARL scenarios without requiring any additional assumptions or mechanisms."**
> >
> > **I don't like this excuse and it is not convincing to me.**
> >
> > We have rewritten the conclusion and added some additional comments regarding the relevance to multi-agent scenarios.
> >
> > “We find that such interpretability is particularly important in multi-agent settings as even small changes in agent performance can lead to large changes in team coordination. Allowing an expert to understand what high-level concepts are used by a policy and reason about how these concepts affect individual agent behavior provides insight into team dynamics, and provides a mechanism for intervening and correcting behavior when necessary.”
> >
> > We have observed such dynamics in the new experimental results shown in Figure 4. In particular, in the right bar plot when we intervene and force both of the defenders to choose the same target this significantly negatively impacts the win rate. This represents (at least some) scenarios where a defender should have chosen a different target, but instead incorrectly chooses the same one as its partner. So a change resulting from a misprediction in a single agent’s policy results in changes to team coordination, and in this case prevents the defenders from effectively stopping the attackers.
> >
> > Given the complex team dynamics that can occur in multi-agent scenarios, we consider any introduction of interpretability into individual agent behavior to be useful.

---

### Author Response · Authors · 2022-08-27
**Updated Manuscript Version and Response to all Reviewers (Part 1)**

Thank you to all the reviewers and the area chair for their thoughtful comments and suggestions. We greatly appreciate the time taken to review our paper.

We have uploaded a new version of our paper to this comment. All changes are shown in blue and we indicate them below.


**Adjusted claims and introduced new experimental results**

In response to Reviewer pdPf’s feedback, we have adjusted our claims such that we no longer claim to directly produce a “decision making rationale”. Instead, we claim that our concept policy models allow a domain expert to reason about a policy in terms of these high-level concepts. Additionally, we removed references to a “general framework” and have reworded text that implies a human expert is capable of intervening and correcting concepts in real-time, instead referring to this as a “domain expert” and in particular, an “oracle”. We recognize that a human expert may be unable to match the intervention performance demonstrated by the algorithmic oracle in our results (where we assume perfect intervention accuracy at every time step), and so use the “domain expert” as a motivating goal but only claim intervention performance with an “oracle”.

To support our above adjusted claim, we have run new experiments and added the results in Figure 4 and the “Behavioral Effects” sub-section. From the paper:

Figure 4 shows the effects of concept values on concept policy model behavior. The left figure visualizes the average position of the defending team as an episode progresses when the Strategy concept is intervened to always take a specific value, showing that the Strategy concept is clearly correlated with the defending team's movement. Specifically, when the attacking team is predicted to move to the left or the right, the defending team moves in the corresponding direction and towards the attackers to intercept. The two bar figures further illustrate behavior effects from concept values; the middle figure shows the probability of a defending agent performing the tag action when the Range concept is intervened and forced to take a value of either 0 or 1, while the right figure shows the resulting win rate when the Target concept is intervened such that both agents are forced to target the same attacking agent rather than predicting their own target independently. These results reveal that high-level concepts can impart meaningful behavioral effects on the downstream policy that align with the concept's interpretation; thus while the concepts do not provide full transparency into the policy's decision making process they still provide the ability for an expert to understand what high-level pieces of information are informing a policy and how these might influence behavior.

**Clarifying comments on the relevance to multi-agent scenarios**

We have rewritten the conclusion and added some additional comments regarding the relevance to multi-agent scenarios.

We find that such interpretability is particularly important in multi-agent settings as even small changes in agent performance can lead to large changes in team coordination. {Clarifying Comment: see the impact to win rate in Figure 4 when a defender chooses the same target as its partner, instead of potentially selecting different targets}. Allowing an expert to understand what high-level concepts are used by a policy and reason about how these concepts affect individual agent behavior provides insight into team dynamics, and provides a mechanism for intervening and correcting behavior when necessary.

---

> ### Author Response · Authors · 2022-08-27
> **Updated Manuscript Version and Response to all Reviewers (Part 2)**
>
> **Additional results regarding the performance of soft concept policy models**
>
> In response to Reviewer EPqJ’s comments regarding the performance of soft concept policy models, we have added the standard deviation to both the Win Rate and Intervened Win Rate columns in Table 1. The high standard deviation for the soft concept policy models shows the increased training instability resulting from the residual layer and accompanying whitening operation.
>
> We note though that the underlying data shows that some seeds yield similar performance to the hard concept policy models – 76% win rate vs 84% win rate for 2v2 and 77% vs 74% for 3v3 for soft and hard models respectively – while other seeds perform worse due to instability during training. We note that the soft concept policy models struggle to learn an effective policy in the 5v5 scenario across all seeds though.
>
> Additional comments in Results section:
>
> The decreased performance in the soft model is due to the fact that it is only trained with a subset of concepts and consequently the residual struggles to reliably encode this information on its own. As evidenced by the large win rate std shown in Table 1 on Lines 1 and 4, some seeds yield similarly strong performance to the hard models while others perform much worse -- with the exception of 5v5 where they failed to learn at all. From this we can conclude that the soft concept policy models can offer comparable performance to hard concept models without requiring a fully-descriptive concept set, at the cost of increased training instability.
>
> **Clarifying comments regarding intervention ablations**
>
> In response to Reviewer EPqJ’s comment regarding the performance of the intervention for the RTOP vs OP real-world model in Table 2, we have performed additional analysis and added clarifying comments on why we believe these results occurred.
>
> In the real-world we observe more pronounced effects due to the differing dynamics, where the strategy concept is more difficult to predict and impacts the behavior more significantly. We conjecture that this is because the agents are less agile in the real-world, and errors in the predicted strategy lead to less-than-optimal positioning. We note that there is an unexpected result here: intervening over every concept but Strategy (Line 3) yields a lower win rate than only Orientation and Position (Line 4). This appears to be a consequence of two factors: intervening over Range and Target but not Strategy increases the prediction error of Strategy which in turn lowers the win rate; and the policy resulting from intervening over only Orientation and Position results in more collisions with other agents during execution, making them easier to tag.
>
> Please see our response to Reviewer EPqJ for further details on this.
>
> **Additional changes to writing clarity and mathematical notation**
> 1. We have adjusted the superscript in Equations 2 and 3 to be (1) and (2) to avoid confusion with exponents.
>
> 2. We have adjusted Equation 2 to remove the conditional probability from $\pi^{(1)}$.
>
> 3. We have added line numbers to Tables 1 and 2 and refer to them in the text, in an attempt to make Section 6 more readable.
>
> 4. We have adjusted the layout of figures and tables to make things more readable, e.g., moving Table 1 and 2 to be close to each other 5. and the Results section where they are discussed.
>
> 6. We have rewritten the conclusion to remove unhelpful sentences and better emphasize our contributions.
>
> 7. We have added clarifying comments on the algorithmic oracle used to perform concept interventions.

---

> ### Author Response · Authors · 2022-08-27
> **Updated Manuscript Version and Response to all Reviewers (Addendum)**
>
> If the reviewers are interested in more information regarding our real world experiments, we have included a video with more details in our supplementary material.

---

### Meta-Review · Area_Chair_TKtr · 2022-08-13

**Recommendation:** Accept (Poster)
**Confidence:** 5

**Metareview:**

# Strenghts
- Interpretability in (multi-agent) reinforcement learning is an important challenge
- Human intervention during the learning process is a promising concept
- The approach was evaluated in a real robot multi-agent scenario.

# Weaknesses
- There are some concerns regarding the novelty of the claimed contributions
- Despite evaluating on physical platforms, the evaluation of the method itself is fairly limited.
- There appears to be a disconnect between the motivation and the actual approach
- Related to the previous point the discussion/interpretation of the experimental results requires additional clarification and justification.

 # Post-Rebuttal Update
The authors addressed the reviewers' concerns very well.
The additional clarifications and relaxed statements/claims improve the paper substantially.
I still ask the authors to make sure that the points further discussed during the rebuttal phase will be addressed in the camera-ready version. Furthermore, the paper will have to be shortened back to 8 pages.


**Best Paper Nomination:**

No